# Thermal and Physico-Chemical Characteristics of Plaster Reinforced with Wheat Straw for Use as Insulating Materials in Building

**Lokmane Saad Azzem *** and **Nadir Bellel**

Energy Physics Laboratory, University Constantine 1, Constantine 25000, Algeria; bellelnadir@umc.edu.dz
* Correspondence: lokmane.saadazzem@gmail.com

**Abstract:** In this paper, a new material consisting of plaster and wheat straw was studied with the purpose of reducing energy consumption. The aim of this study is to test this new compound for use as an insulation material in buildings, where the samples were prepared by mixing wheat straw after grinding it in different proportions from 0% to 15%. On the other hand, the physico-chemical properties and thermal conductivity of the samples were experimentally investigated, and the time lag and energy savings for the samples were also studied. The results showed that the addition of wheat straw leads to an increase in the time lag and also to a decrease in the thermal conductivity, which leads to an improvement in the thermal resistance and energy savings. As well, fiber addition has no effect on the chemical composition of the matrix, as shown by FTIR and XRD analyses.The findings of the DSC and TGA analysis indicate that the inclusion of wheat straw fibers has an effect on the thermal characteristics of the matrix. This new biocomposite can be used as an additive to plaster to create environmentally friendly composite materials for thermal insulation in buildings.

**Keywords:** wheat straw/plaster composite; physico-chemical; thermal conductivity; energy saving

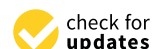

## 1. Introduction

Demographic growth and economic and technological progress in the world have led to excessive energy consumption [1]. The construction sector is among the sectors that consume energy at a global level, approximately 40% of global consumption [2]. On the other hand, in Algeria, construction is a major consumer of energy, accounting for 42% of primary energy [3]. The use of operational energy in cooling and heating increases energy consumption and emissions, therefore researchers aim to decrease energy usage, emissions, and waste [4]. Building insulation is often made using materials derived from petrochemicals (mostly polystyrene) or from natural sources treated with substantial energy consumption (glass and stone wool). These materials have a significant negative impact on the environment, primarily due to the production stage, which includes the use of non-renewable materials and the consumption of fossil fuels, and the disposal stage, which includes difficulties in reusing or recycling the products at the end of their lives [5,6]. The researchers focus on developing the building so that it becomes environmentally friendly, and therefore the materials manufactured for insulation become undesirable, so they are turning to materials from natural sources. Plaster was one of the first construction materials invented. plaster boards are widely used for interior walls and ceilings due to their ease of manufacture, environmental friendliness, beauty, low cost, and high fire resistant capabilities [7].

Natural fibers have lately gained popularity as an alternative component for composite materials among researchers, engineers, and scientists. Because of their cheap cost, relatively excellent mechanical qualities, high specific strength, non-abrasive, ecofriendly, and bio-degradability, they are being used as an alternative to traditional fibers such as glass, aramid, and carbon [8,9]. As stated in [5], who shows the thermal conductivity of

different natural materials that can be used to increase the insulation in the building. In general, the addition of natural fibers led to an increase in the thermal insulation properties, as Ashour et al. [10], as well as Braiek et al. [11], studied the use of a group of different natural fibers as additional materials for gypsum, where the results confirmed that adding these fibers leads to an improvement in thermal insulation. These results were supported by Lamrani et al. [12], who used peanut shells as an additive in gypsum. In addition, according to the literature [13–16], who studied the addition of natural- source materials to the gypsum or plaster. The incorporation of these materials results in an increase in the thermal insulation of the construction.

In the building envelop, controlling heat transfer, ensuring thermal comfort, and saving energy are all critical factors. Insulation must perform well throughout the building's life cycle. Also, certain research [17,18] verified the utilization of natural fibers in gypsum to create novel compounds that have satisfactory physical properties for use in the construction sector.

When deciding which natural materials to incorporate, the physico-chemical characterisation of compounds containing natural fibers is critical. Physical and chemical qualities, as well as thermal parameters, should all be taken into account.

The present paper aims to improve the thermal performance of plaster material by incorporating different mass fractions of wheat straw. The new composite construction material makes it possible to meet the requirements of the building's thermal comfort while reducing energy consumption and ensuring a healthy environment by minimizing greenhouse gas emissions. Experimental characterization of thermal properties of the composite material was performed to identify apparent density, thermal conductivity, thermal diffusivity, thermal effusivity, and volumetric thermal capacity. The physicochemical characterization of this composite was also analyzed, in addition to the thermophysical characterization. The specific objective of this work is to provide a comprehensive overview of the main findings related to the use of wheat straw in plaster.

## 2. Materials and Experimental Methods

### 2.1. Materials

#### 2.1.1. Wheat Straw

In this work, the wheat straw was collected from (Setif-Algeria) by removing the stalk and cutting it into small pieces. It is dried by placing it in a 60 °C oven in the laboratory for 24 h and crushing it using an electric grinder (Broyeur à fléaux SK 100 comfort) into small pieces. The length of wheat straw fibers is between 0.5 and 3 mm. The choice of this part of the spike was made for its thermophysical properties [5].

#### 2.1.2. Plaster

In this work, the plaster produced by Al-Taouab Company (Algiers-Algeria) was selected, and it mainly consists of calcium sulfate ($CaSO_4.\frac{1}{2} H_2O$) and is used for indoor wall coatings of buildings and roof ceilings. It is obtained by the thermal drying of gypsum (calcium sulfate dihydrate) at a certain temperature, and this is summarized in the following Equation (1):

$$CaSO_4.2H_2O + Heat \rightarrow CaSO_4.\frac{1}{2}H_2O + \frac{3}{2}H_2O \tag{1}$$

### 2.2. Samples Preparation

The composite samples were formed by mixing plaster with water and adding wheat straw in the following proportions: 5%, 10%, and 15%. Before being mixed with the other components of the composite, the wheat straw is ground in a milling machine (Broyeurs à fléaux SK 100 comfort) from the company Retsch located in (Retsch-Allee 1-542781 Haan). The samples are prepared by mixing plaster with wheat straw, where the wheat straw is added in the proportions mentioned above, in order to conduct physico-chemical analyses. As for the thermal conductivity analysis, wheat straw was combined with the plaster in

quantities of 5%, 10%, and 15% and then mixed together. The mixture is constantly mixed until well mixed. Mixing plaster with wheat straw in the aforementioned proportions is necessary before adding water to obtain homogeneity in the mixture, Then water is added to the mixture in which the plaster was mixed with wheat straw in the previously mentioned proportions, as the ratio of water to plaster is 0.7. The mixture is constantly mixed until a dough is obtained. It was put in molds for 72 h before being removed. They are then taken from the molds and stored in a laboratory for 28 days at room temperature. The thermal conductivity analysis, on the other hand, is carried out by creating samples with dimensions of 16 cm × 8 cm × 5 cm. In Table 1, the nomenclature for each sample is shown. Figure 1 shows composite samples constructed from plaster reinforced with wheat straw fibers.

**Table 1.** The code and composite samples.

| Refrences | Sample Composite |
| --- | --- |
| CP | Comerciale plaster |
| CP5 | Comerciale plaster + 5% of Wheat straw |
| CP10 | Comerciale plaster + 10% of Wheat straw |
| CP15 | Comerciale plaster + 15% of Wheat straw |

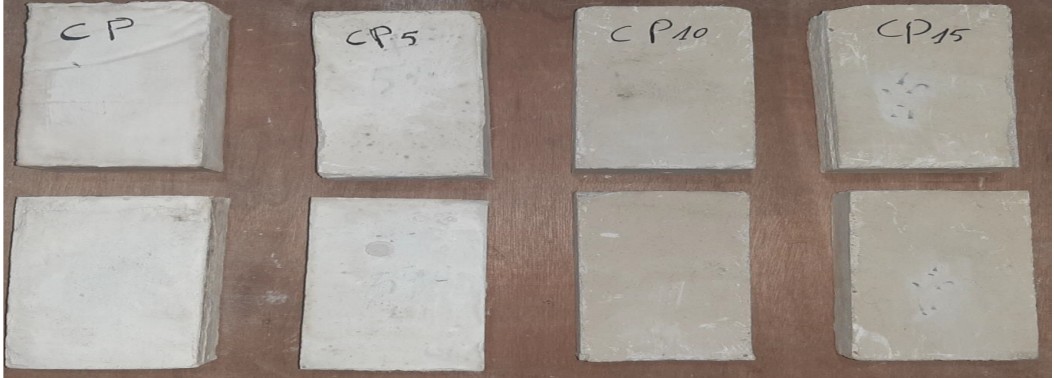

**Figure 1.** Picture of composite sample.

### 3. Characterization

*3.1. Physico-Chemical Characterization of Materials*

3.1.1. TGA

In this study, SETARAM Labsys Evo-gas was used to perform Thermogravimetric analysis (TGA), with the option of Simultaneous TGA/DTA, for studying thermal degradation of the prepared materials. The samples are heated at a rate of 10 °C/min in a nitrogen environment with a flow rate of 40 mL/min from 50 °C to 1200 °C.

3.1.2. DSC

Differential scanning calorimetry (DSC) analyses were performed to investigate the thermal characteristics of samples generated using a DSC131 Evo device in a nitrogen environment in a temperature range of 40 to 700 °C, with a flow rate of 40 mL/min and a heating rate of 10 °C/min.

3.1.3. XRD

Analyses of X-ray diffraction (XRD) were carried out using a copper anode and radiation K1 at wavelength 1.5406 Å, produced in the range 2θ = 5–100° with a step size of 0.025 at 40 Kv and 25 mA. using a BRUKER X-ray diffraction D8 ADVANCE A25. In order to study the analysis of crystalline and mineral phases of gypsum plaster and the effect of adding wheat straw to plaster.

### 3.1.4. FTIR

FT-IR analysis was performed to investigate the impacts of wheat straw added to plaster, using the JASCO FT/IR-6300 equipment in the 4000–400 cm$^{-1}$ spectral region with a resolution of 2 cm$^{-1}$.

### 3.1.5. SEM

The morphology of wheat straw and plaster was studied with the use of a scanning electron microscope (SEM) MEB Quanta 250 from the FEI (Field Electron and Ion Company), Headquartered in Hillsboro, Oregon, USA. in order to study the effect of adding wheat straw to plaster and also the microstructure of the samples and pore geometry, as well as the homogeneity of the two materials.

### 3.2. Thermophysical Characteristics Measurements
Apparent Density

The construction sector is interested in lightweight materials, which gives an economic incentive. As a result, determining apparent density becomes a crucial task for each study. To calculate the apparent density, the sample's weight and dimensions must be measured. The following Equation (2) may be used to compute the bulk density:

$$\rho = \frac{M}{V} \tag{2}$$

where: $M$ and $V$ are respectively the mass and the volume of the samples.

### 3.3. Thermal Conductivity

The thermal conductivity of the composite samples is measured using the CT-meter equipment in accordance with NFISO 8894-1 2nd edition 15/05/2010. The measuring methodology is based on the hot wire method and allows for the determination of a material's thermal conductivity based on the temperature fluctuation detected by a thermocouple positioned near a resistive wire. The probe is made up of resistive wire and a thermocouple in an insulating kapton support that is sandwiched between two samples of the substance to be evaluated. The user determines the heating duration based on the substance to be tested and the type of sensor utilized. The outcomes are presented on the device's screen. The accuracy of this setup is 5%, the temperature range of measurement test is from 20 to 30 °C for thermal conductivity materials from 0.01 to 10 W.m$^{-1}$.

$$R_{th} = \frac{t}{\lambda} \tag{3}$$

where: $t$ is the thickness of the sample and $\lambda$ is the thermal conductivity of the sample.

### 3.4. Time Lag

The period of time required for a heat wave to spread from the outer surface to the inner surface of a wall is known as the time lag. Time lag is a key factor in determining the heat storage capacities of any material. And since the change in time lag is linked to the thermal diffusivity of materials ($a = \frac{\lambda}{\rho C_p}$) and both depend on the properties, $\lambda$, $\rho$, and $C_p$, which are all of them are the thermal conductivity of material, the density of material, and the specific heat of material, respectively [19]. And as it has been studied in the literature [19,20], The outer surface temperature of a building wall can be represented by a series of sinusoidal components. Furthermore, the wall might be seen as a semi-infinite body. In this example, Equation (4) gives the time lag:

$$T_{lag} = \frac{Tt}{2\pi} \times \sqrt{\frac{\pi \rho C_P}{\lambda T}} \tag{4}$$

where: $T$ is the periodic cycle of temperature variation (h), $k$ thermal conductivity ($Wm^{-1}K^{-1}$), density ($kg\,m^{-3}$) and $C_p$ specific heat ($Whkg^{-1}K^{-1}$), $t$ is the material thickness (m). The following is the equation when the temperature cycle change is 24 h:

$$T_{lag} = 1.38t \times \sqrt{\frac{\rho C_p}{\lambda}} \tag{5}$$

*3.5. Energy Saving*

All studies conducted on these samples were for the purpose of minimizing heat loss and obtaining better performance in terms of thermal insulation, so the energy gained before and after adding wheat straw was calculated. These studies and experiments were conducted to improve performance in terms of the insulation properties of this composite material made of wheat straw and gypsum for use in the construction field, so a comparison was made between two external walls of different compositions, one of which contained the composite material and the other of which did not [21]. Hence, the heat flux for both walls is:

$$\phi_{plaster} = \frac{\lambda_{plaster} a \Delta T}{t} \tag{6}$$

$\lambda_{plaster}$ = the thermal conductivity of plaster.
a = the area of the wall.
$\Delta T$ = temperature variation.
t = the thickness of the wall.

$$\phi_{comp} = \frac{\lambda_{comp} a \Delta T}{t} \tag{7}$$

The heat flux across the wall that holds the composite material is known as $\phi_{comp}$. The heat flux across a pure gypsum wall is denoted by $\phi_{plaster}$. The conductivity's of composite and pure gypsum materials are represented by $\lambda_{comp}$ and $\lambda_{plaster}$, respectively. And for an area of 1 square meter (a = 1 $m^2$) :

$$\phi_{plaster} = \frac{\lambda_{plaster} \Delta T}{t} \tag{8}$$

$$\phi_{comp} = \frac{\lambda_{comp} \Delta T}{t} \tag{9}$$

And also, for the same thickness:

$$\frac{\phi_{comp}}{\phi_{plaster}} = \frac{\lambda_{comp}}{\lambda_{plaster}} \tag{10}$$

Through the previous equations, the energy savings is calculated as follows:

$$E_s aving = 100 \times \left(1 - \frac{\phi_{comp}}{\phi_{plaster}}\right) \tag{11}$$

## 4. Results

*4.1. Thermal Property*

4.1.1. TGA

Figure 2 shows the results obtained for pure plaster and after adding wheat straw in different proportions. The weight loss of the analyzed samples follows a similar pattern, which is shown in the curve Figure 2. The difference lies in the proportion of weight loss, which varies based on the amount of wheat straw used in the plaster, such that the greater the percentage of wheat straw added in the samples The weight loss was greater compared to the plaster sample. Weight loss is divided into four sections on each curve, which can

easily be recognised, as shown in Figure 3. In the first region, the weight loss in which the temperature is less than 120 °C is due to the water adsorbed in relation to the plaster and to the subtraction of moisture content for wheat straw [22].

While in the second region, the weight loss returns to the chemical bonding of water from the aqueous salts of plaster and to the deterioration of the volatile matter in the wheat straw, and it is generally in the temperature range of 120 to 360 °C [23,24]. The third region has a temperature range of 360 to 650 °C, the weight loss in this region is connected to the chemical interaction of water with hydraulic compounds in plaster and the breakdown of carbon in wheat straw. In the last thermal range, which is at a temperature of more than 650 °C, the weight loss is due to the loss of carbon dioxide, which is produced during the degradation of carbonates for plaster, and also to the production of ash for wheat straw [25].

From the results that show the difference in mass loss between the pure plaster sample and the samples added to wheat straw, it is clear that the thermal characteristics of the compounds are affected by the addition of wheat straw.

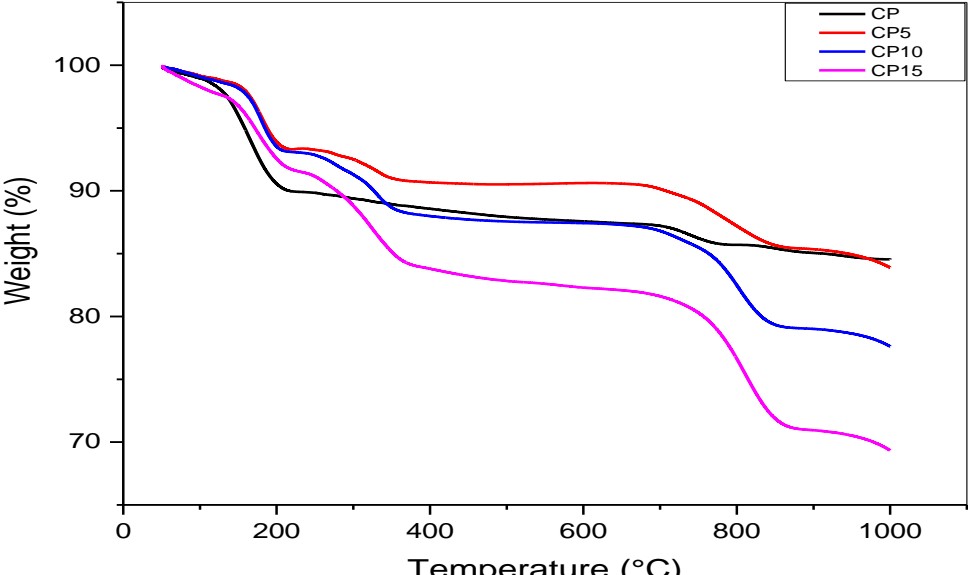

**Figure 2.** TGA of all samples.

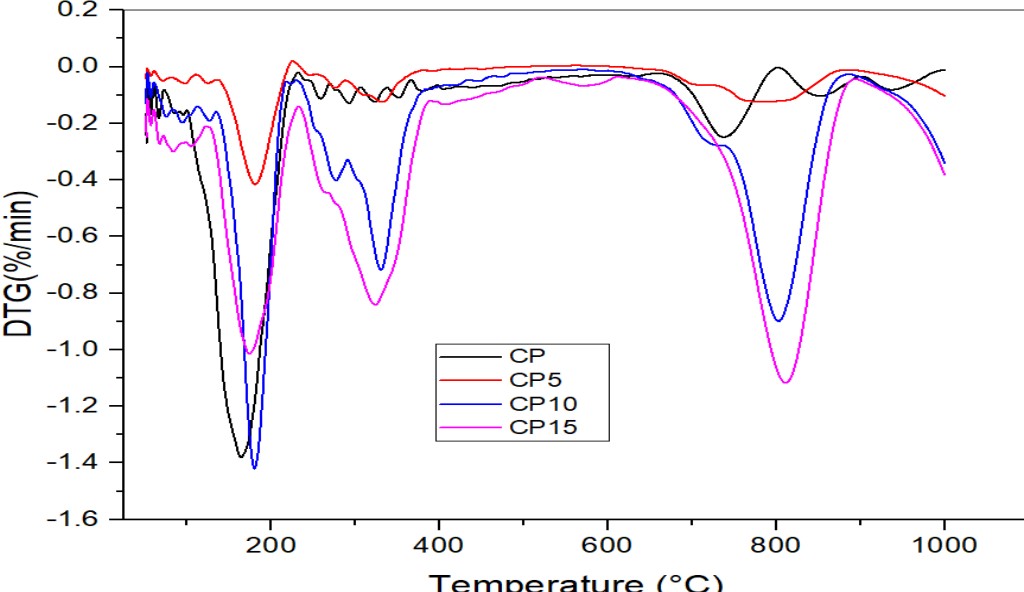

**Figure 3.** DTG of all samples.

### 4.1.2. DSC

From Figure 4 that shows the DSC curve, there are three peaks for all the composite samples, where the temperature of the peaks changes. When the proportion of wheat straw is increased and added to the plaster, the temperature increases in the first peak, which is an endothermic peak, where the temperature is between 70 and 75 °C. This peak corresponds to the evaporation of water at the heating of the samples [26,27]. The second peak, which has the most heat flux, is also an endothermic peak, where the temperature is between 150 and 170 °C, which corresponds to a direct conversion of Sodium Sulfate Dehydrate to Calcium Sulfate Anhydrite III [27,28]. Furthermore, an exothermic peak between 350 and 370 °C was detected, which corresponds to the transition of soluble calcium sulfate anhydrite III to insoluble calcium sulfate anhydrite II, in addition to the thermal decomposition of wheat straw [26,29]. The difference in heat flow is due to an increase in the amount of wheat straw in the plaster, which necessitates a high heat flux in order for the wheat straw to thermally decompose. All curves in Figure 4 are nearly identical. The slight difference in the curves compared to the plaster curve is due to the thermal decomposition of wheat straw added to the plaster, indicating that no new products or chemicals were detected.

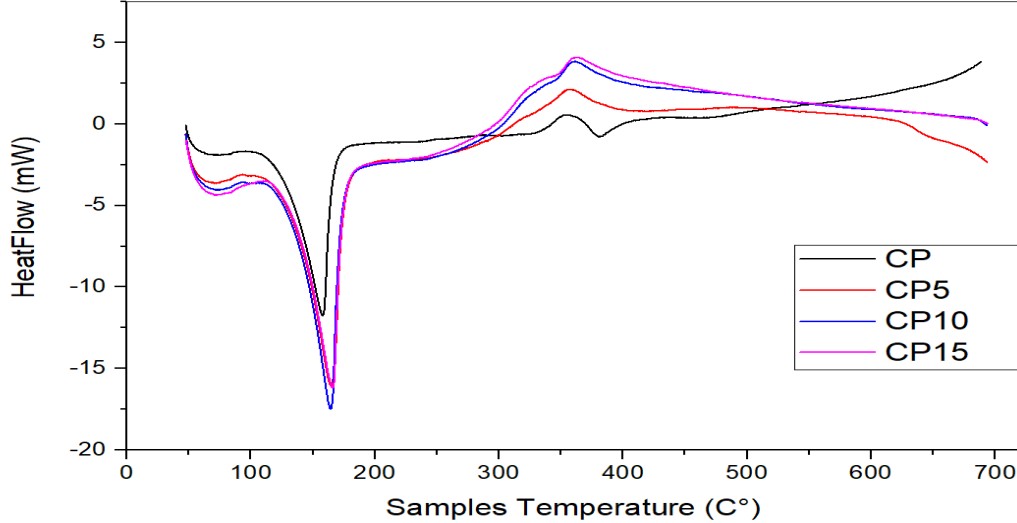

**Figure 4.** DSC of all samples.

### 4.2. Microstructure

### 4.2.1. XRD

Figure 5 shows the results of the DRX analysis of the sample of plaster, where the following peaks were obtained: $CaSO_4 \cdot \frac{1}{2} H_2O$ represents calcium sulfate hemidrite, also known as basanite, as well as dihydrate $SO_4 \cdot 2H_2O$ and anhydrite $CaSO_4$, all of which are metal phases as mentioned in the literature [30,31]. And through the other Figure 6, which also show the analyzed DRX of the samples in which different percentages of wheat straw were added, as in all the results shown in Figure 6, the gypsum minerals Crystalline is the main product in all compounds. It is clear that there is no change in the mineral composition of the materials and also the absence of new chemical elements, which means that adding wheat straw does not affect the crystalline shape of the matrix [31].

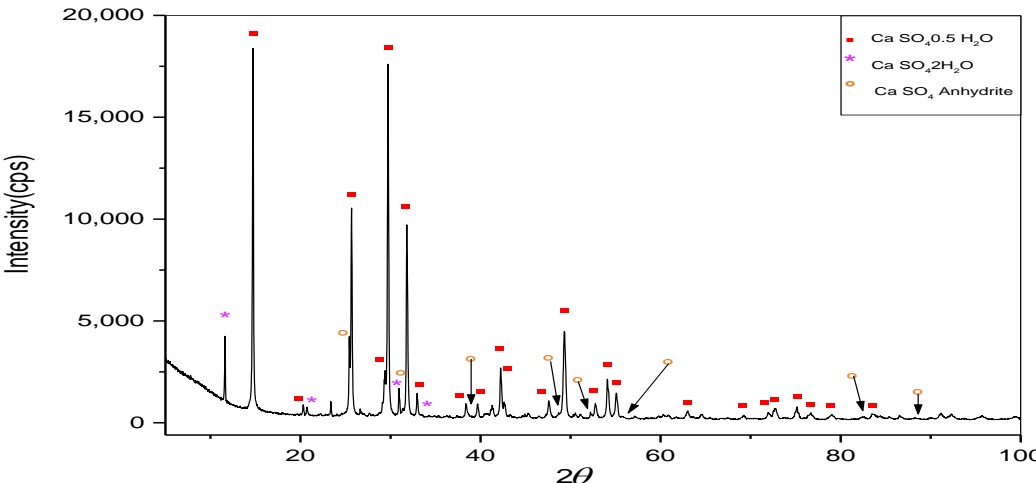

**Figure 5.** XRD of the pure plaster.

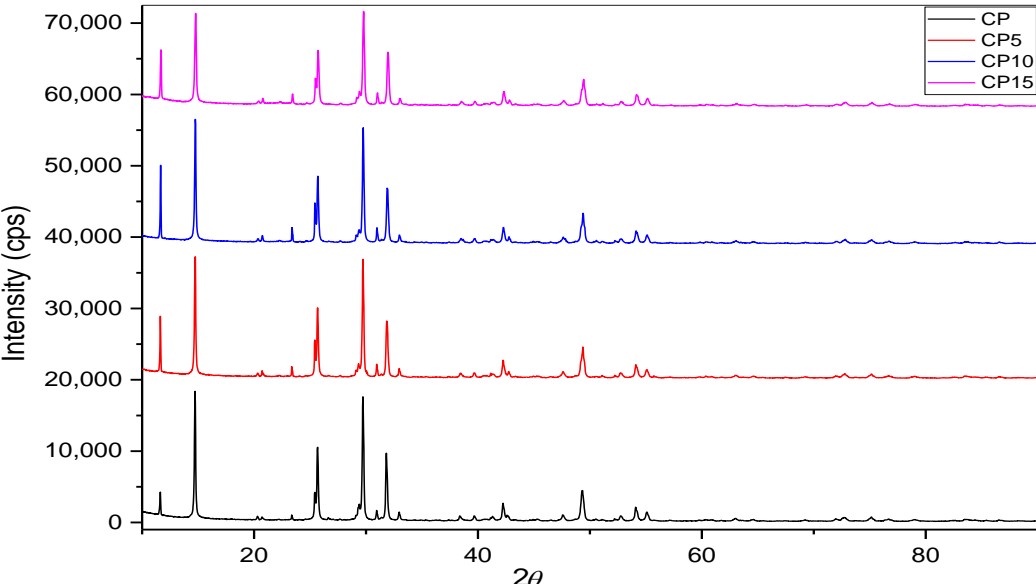

**Figure 6.** XRD of all samples.

4.2.2. FTIR

Figure 7 shows the FTIR analyses of the wheat straw and the plaster. Wheat straw fibers were analyzed to determine the chemical constituents. Figure 7a shows the IR of the bio-based wheat straw. The peaks in the 3330 cm$^{-1}$ area are created by the O-H stretching vibration, which is induced by the vibration of the hydrogen connected hydroxyl O-H group [32,33]. As with banana fiber, the aliphatic saturated C-H stretching vibration in cellulose and hemicellulose fibers causes the peaks at 2917 cm$^{-1}$ [32–35]. The peak of 1731 cm$^{-1}$ is formed by the ester linkage of acetyl hemicellulose and uronic ester groups or the ester linkage of lignin's carboxylic frulic groups and p-coumaric acids [32–36]. The reect C-H asymmetric deformation is responsible for the peak of 1370 cm$^{-1}$ [32,35,37]. The peaks of 1509 cm$^{-1}$ and 1425 cm$^{-1}$ are produced by the aromatic C=C stretch of lignin aromatic rings [35,36,38]. The C-O stretch and deformation bands in cellulose, lignin, and residual hemicellulose range from 1200 to 1056 cm$^{-1}$ [32,34,35,39]. The peak of 1033 cm$^{-1}$ in hemicelluloses is caused by C-O, C-C stretching, or C- OH bending [40,41]. the peak of 903 cm$^{-1}$ owing to the $\beta$-glycosidic connections of the cellulose glucose ring [36,38,41]. Analysis of the FTIR results for plaster is shown in Figure 7b, The stretching vibration bands of O-H characterize the two bands at 3605 and 3555 cm$^{-1}$, respectively [42,43]. The 1618 cm$^{-1}$ peak represents a strong water molecule water anion characterized by

the bending vibration of O-H [42]. Peaks 1140,1111,1087 cm$^{-1}$ indicate the asymmetric stretching vibration of SO$_4$ tetrahedra [44], whereas a minor peak at 1005 cm$^{-1}$ represents the symmetric stretching vibration of SO$_4$ tetrahedra [44,45], the two peaks at 659 and 594 cm$^{-1}$ also represent the asymmetric bending vibrations of the SO$_4$ tetrahedron [41]. The Figure 8 shows Analysis results FTIR for plaster samples after adding wheat straw, for each of the results of analyzing natural plaster before and after adding wheat straw fibers, there is no difference or change in the peaks in the plaster, which means that there is no chemical reaction between the plaster Wheat straw and this indicates that the properties of plaster do not change after adding wheat straw, and therefore the two components can be combined with each other in the binder without any risk of chemical decomposition.

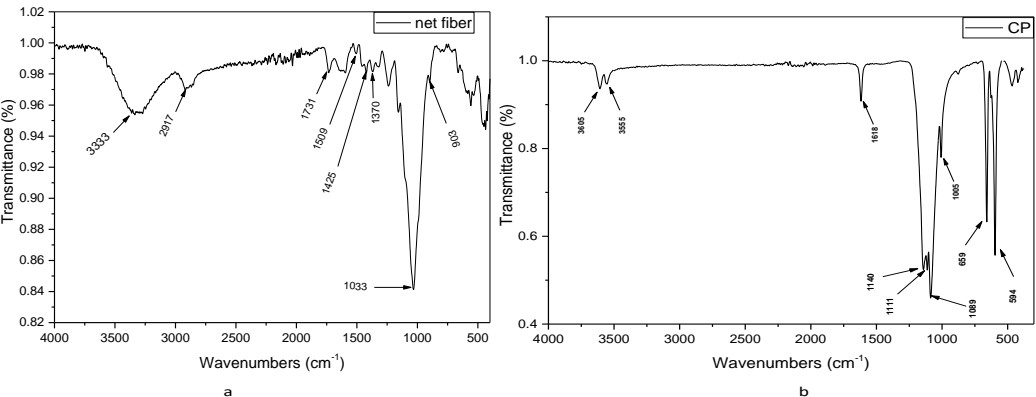

**Figure 7.** FTIR of: (**a**) Net Fiber; (**b**) plaster net.

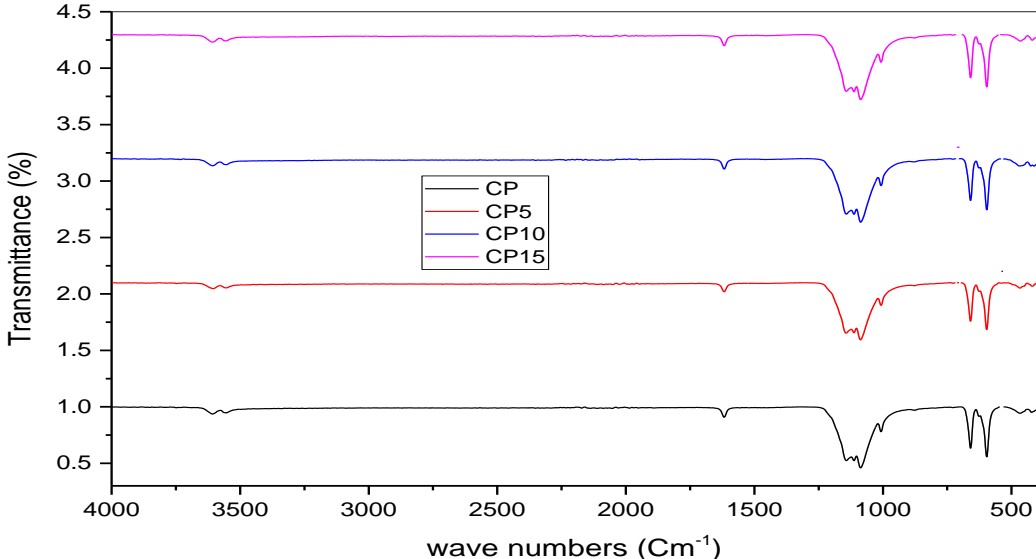

**Figure 8.** FTIR spectra of plasters before mixing and after mixing.

### 4.2.3. SEM

　　Figure 9a shows the image of wheat straw fibers, which are cylindrical in shape with some threads, cells, and pores that allow them to adhere well to plaster. Figure 9b shows the picture of wheat straw fibers adhered to the plaster, and this is evident when comparing pure plaster and samples to which wheat straw has been added.

　　Figure 10 shows the discrepancy between the pure plaster sample and the other samples, as the addition of wheat straw resulted in a greater number of pores that had been formed when preparing the samples. The wheat straw fibers absorb a significant amount of water during preparation, which leads to a loss of the amount of water absorbed after the

samples are dried. In addition, increased pore size results in increased thermal insulation, and this result corresponds to [20].

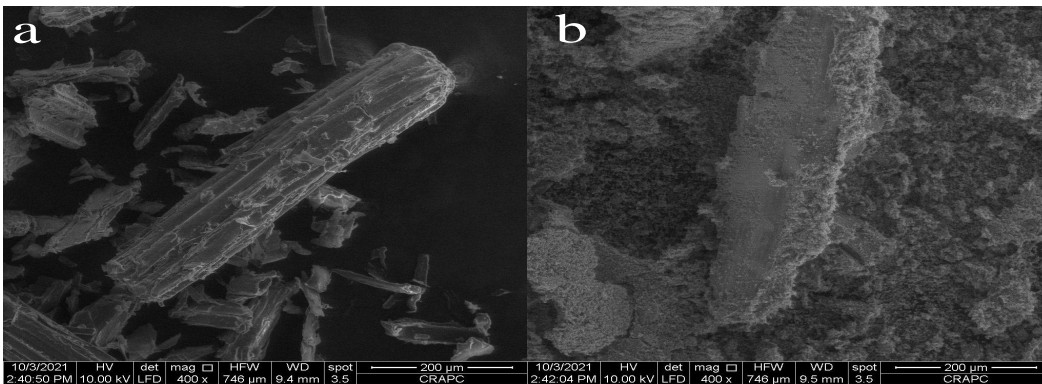

**Figure 9.** (**a**) net fiber form; (**b**) plaster sticking to wheat straw fibers.

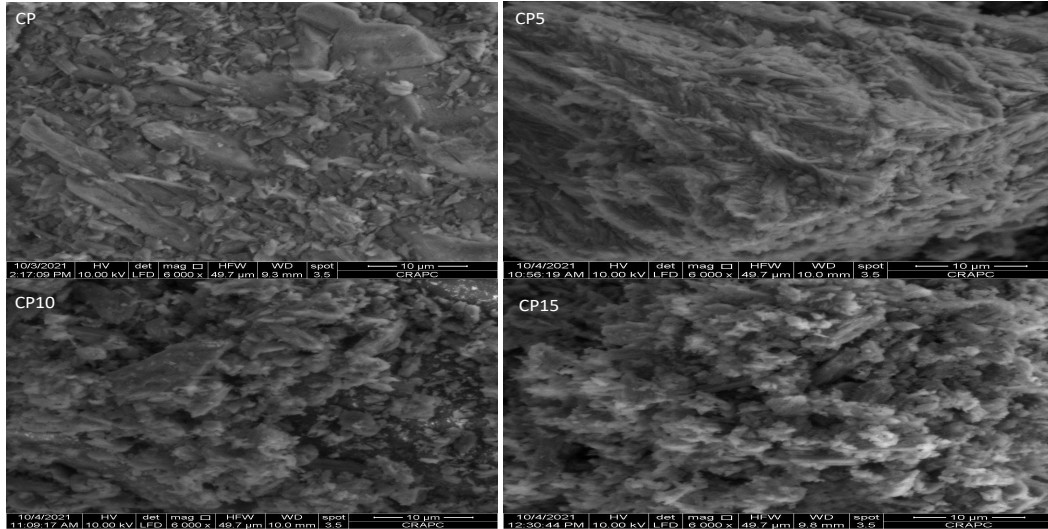

**Figure 10.** Scanning electron microscope images of the binder specimens.

*4.3. Thermophysical Properties*

4.3.1. The Apparent Density

From the results in Figure 11, it can be seen that the bulk density has an inverse relationship with the percentages of wheat straw added to the samples, where the bulk density decreased from 1103.13 kg.m$^{-3}$ for pure plaster to 1043.75 kg.m$^{-3}$ for the sample containing 15% wheat straw. The lightness of the sample containing 15% wheat straw was 5.38% compared to the plaster. This is due to the pores created by adding wheat straw as well as the low density of wheat straw [10]. Furthermore, given a specific weight ratio, the volumes filled by the wheat straw were substantially higher than those occupied by the plaster. In general, these findings show that increasing the amount of wheat straw additions in the mixture reduces the density of the sample matrix [46].

4.3.2. Thermal Conductivity

Thermal conductivity is one of the most important characteristics of materials used in building walls. The change in thermal conductivity as a function of fiber concentration is seen in Figure 12. It shows a decrease in thermal conductivity as the wheat straw content increases, where the thermal conductivity of the plaster sample is 0.408 W/m.K, and It rapidly diminishes as the proportion of wheat straw increases in the samples until it reaches 0.324 W/m.K for the sample that contains 15% wheat straw, where the percentage decreases

thermal conductivity by about 20.6%. Similar results were achieved in [10], which the thermal conductivity of wheat straw as fibers added to gypsum was discussed.

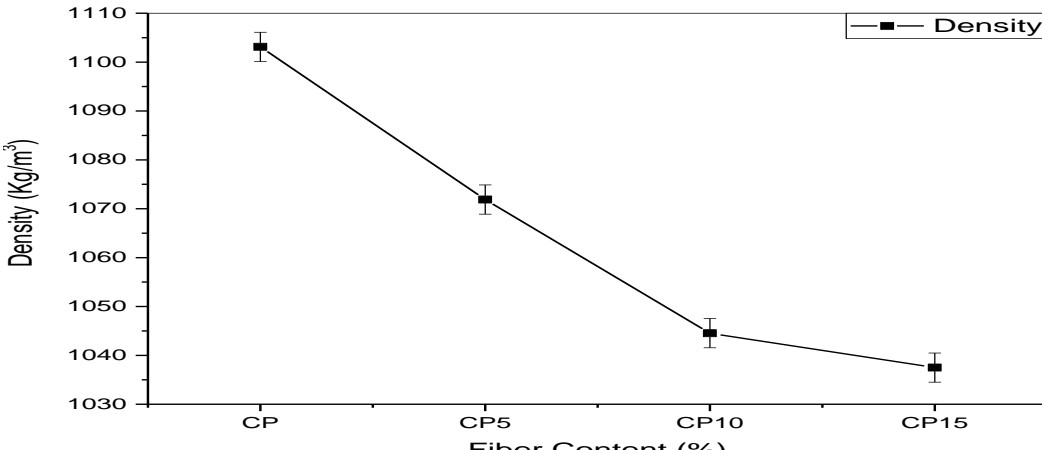

**Figure 11.** Density variation versus wheat straw of composites.

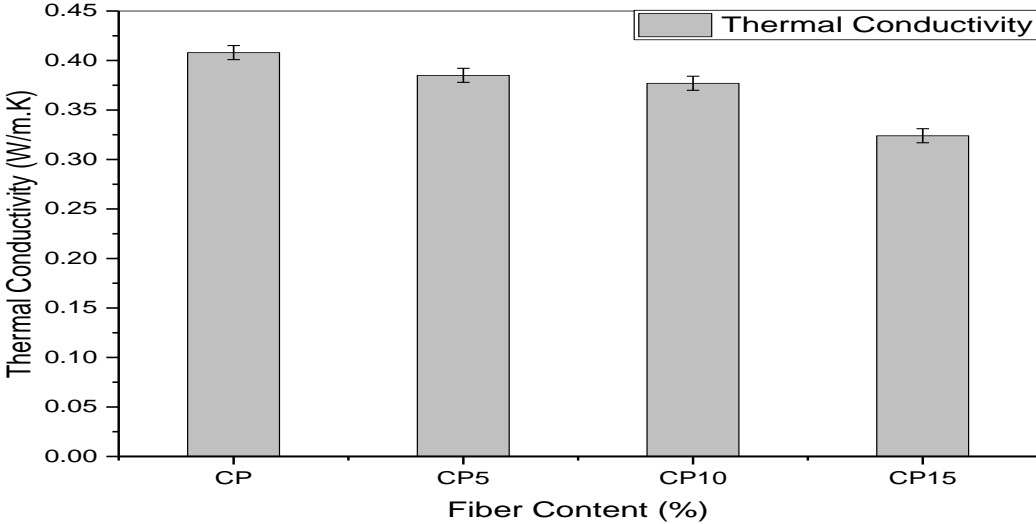

**Figure 12.** Thermal conductivity and wheat straw percentage relation in the specimens.

These findings can be explained by a decrease in the density of the tested samples, which is due to an increase in the percentage of wheat straw whose density is less than that of plaster samples, and on the other hand, by a gradual increase in the amount of wheat straw with low thermal conductivity when compared to plaster. Natural fibers, in general, cause a reduction in density and the formation of porosity in samples. As a result, the formation of pores results in the presence of air, whose thermal conductivity is reported to be 0.026 W/m.K, which is low. Similar outcomes were obtained by [20]. On the other hand, as shown in Figure 13 and the findings in Table 2 thermal diffusivity reduced from $3.4 \times 10^{-7}$ to $2.87 \times 10^{-7}$, representing a 15.6% decrease in heat transfer. The volumetric heat capacity likewise decreased by 5.84 %. This is what was studied in [47].

### 4.3.3. Time Lag

The graphs indicate the time delay changes in terms of thermal energy and thermal diffusion. where the thickness of the sample was adopted as 5 cm. Through the Figures 14 and 15, As comparable results were revealed, there is an inverse connection between the time lag and that of thermal energy and thermal diffusion, which means that the greater the values of thermal energy and thermal diffusion, the lower the value of the time lag [19,20].

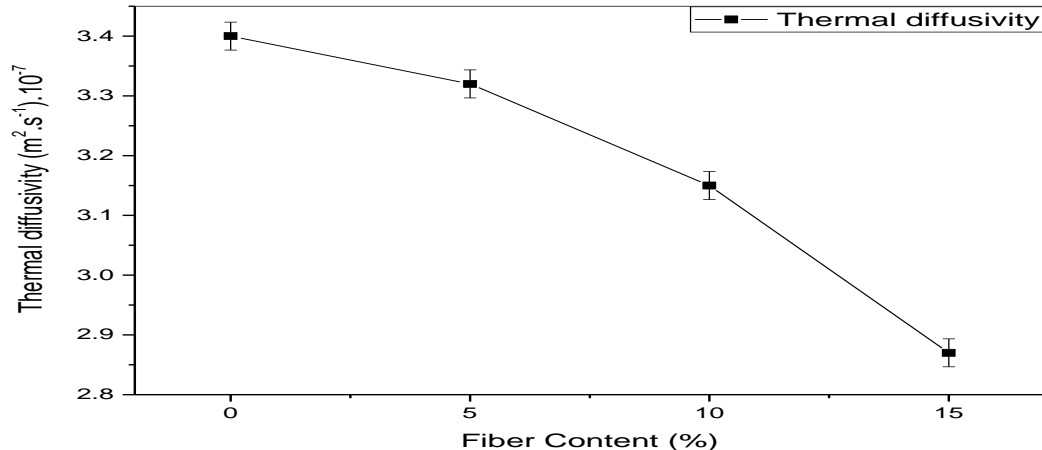

**Figure 13.** Thermal diffusion in terms of fiber content.

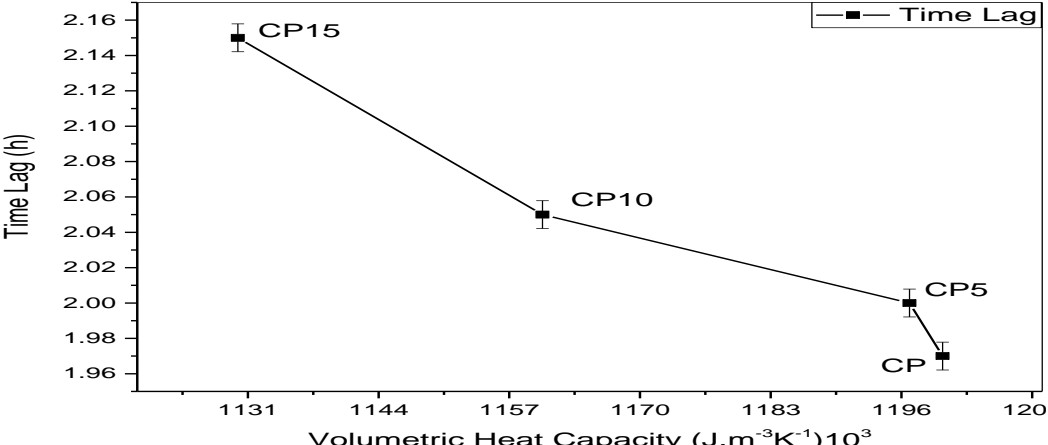

**Figure 14.** Time lag as function of Thermal energy.

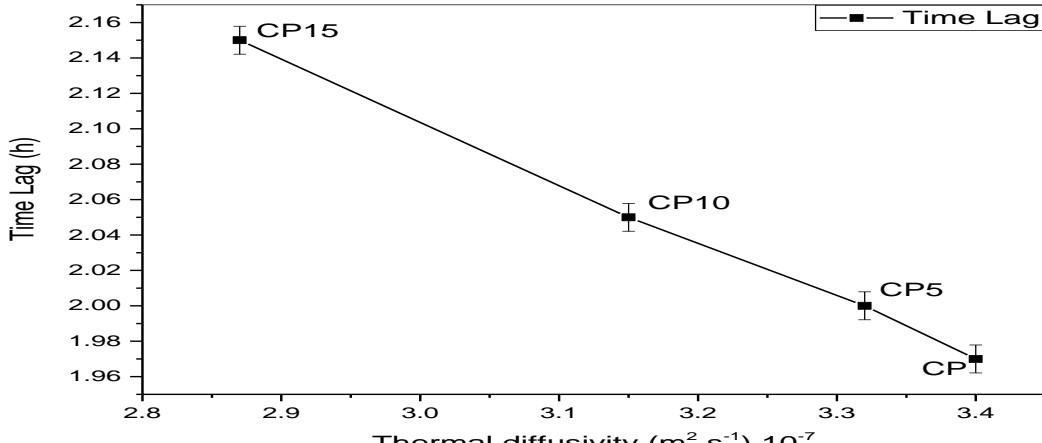

**Figure 15.** Time lag as function of thermal diffusivity.

It was discovered in Table 2 that increasing the proportion of wheat straw in the plaster samples resulted in an increase in thermal resistance as well as an increase in the time lag of the plaster samples; whereby, with the incorporation of 15% of wheat straw, it led to an improvement of 9.14% compared to the plaster.

**Table 2.** Thermal properties of composites materials.

| Samples | $\lambda$ $W\ m^{-1}K^{-1}$ | $R_{th}$ $(W^{-1}\ m^2\ K)$ | $\rho C_p$ $(Jm^{-3}\ K^{-1})10^3$ | a $(m^2\ s^{-1})10^{-7}$ | Time Lag (h) |
|---|---|---|---|---|---|
| CP | 0.408 | 0.123 | 1200.1 | 3.4 | 1.97 |
| CP5 | 0.385 | 0.13 | 1160.3 | 3.32 | 2 |
| CP10 | 0.377 | 0.133 | 1196.8 | 3.15 | 2.05 |
| CP15 | 0.324 | 0.154 | 1130.0 | 2.87 | 2.15 |

As for Figure 16, in general, the time lag increases with increasing thermal resistance. That is, there is a direct relationship between the time lag and the thermal resistance, and these results show behavior similar to that presented in [48,49].

On the other hand, the thickness has an impact on the time lag. It is through Figure 17 that shows the impact of thickness on the time lag of the prepared compounds, and as shown, this is not surprising because as the wall thickness gets thicker, its heat storage capability increases, and this can be explained by the fact that the wall, which has a small thickness, the heat wave spreads from the outside to the inside the wall without any delay. Furthermore, if the wall is made of insulating materials with a low heat capacity and thermal conductivity, the value of time lag increases, which is confirmed by [50], who studied such cases, so that the time lag values are small when the thickness is less than 10 cm. After this thickness, the values of the time lags start to increase.

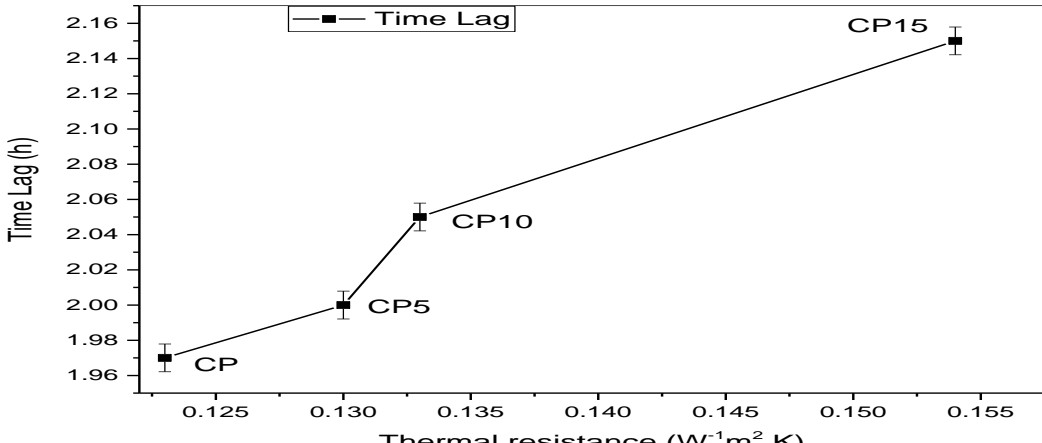

**Figure 16.** Time lag variation versus thermal resistance of composites.

According to the findings, thermal resistance is not the only factor influencing time lag; it is also affected by a complex interplay of material density, specific heat capacity, thickness, and thermal conductivity, as well as the efficiency of insulation materials in holding back heat. This finding is consistent with the findings of [19,49,51], who discovered that the time lag of a wall with varied configurations is impacted by the thermophysical parameters of the wall's material, thickness, and orientation.

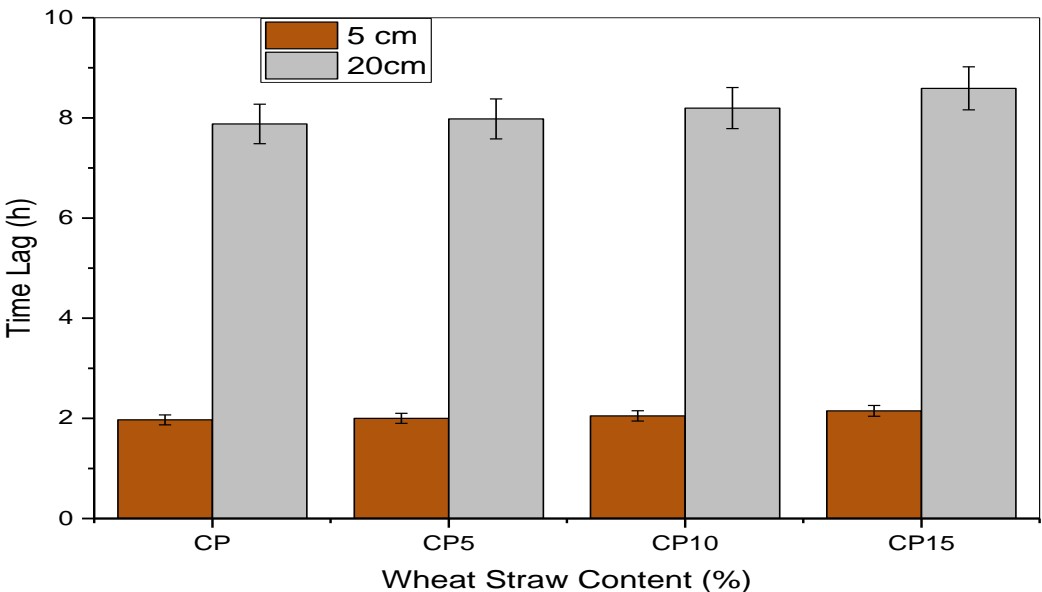

**Figure 17.** variation of the time lag as function of the thickness.

### 4.3.4. Energy Saving

Through the Table 3 that provides results for some compounds in which natural fibers are incorporated which are used in thermal insulation. According to the table, it can be noted that in this study, Plaster containing 15% wheat straw has good thermal performance, i.e., a small thermal conductivity and better performance in terms of energy savings compared to plaster containing 25% wheat straw (dry basis) which was studied by [10]. And by an other comparison with results of [10], the composite of plaster and Wood shavings has a small thermal conductivity and energy savings (dry basis) of 19.5% compared to pure plaster. On the other hand, the compound of plaster and 15% wheat straw showed a greater thermal conductivity than that of plaster and Wood shavings, and this is due to the proportion of incorporated natural fibers, while it showed an acceptable percentage of energy savings with 20.6% compared to the plaster, which allows it to be proposed as one of the materials for thermal insulation.

**Table 3.** Wheat straw/plaster composite compares to other construction materials in terms of thermal conductivity and energy savings.

| Materials | $\lambda$ ( $Wm^{-1} K^{-1}$ ) | Energy Saving % | References |
|---|---|---|---|
| Net plaster | 0.408 | 0 | This work |
| Wheat straw /Plaster (15%) | 0.324 | 20.6 | This work |
| Wood shavings gypsum (40%) | 0.2 | 18.8 | [46] |
| Wheat fiber/plaster (25%) (dry basis) | 0.33 | 4.3 | [10] |
| Barley fiber/plaster (25%) (dry basis) | 0.29 | 18.8 | [10] |
| Wood shavings / plaster (25%) (dry basis) | 0.28 | 19.5 | [10] |

## 5. Conclusions

In this study, the integration of biosourced wheat straw with plaster was researched, where the physicochemical characteristics as well as the thermal properties of the new compound were studied, and the main results were stated as follows:

1. The FTIR and DRX results revealed that after incorporating varying amounts of wheat straw into the plaster, there was no influence on the level of chemical characteristics and that it was chemically stable, as well as no changes at the level of the matrix's microstructure. On the other hand, TGA results showed strong thermal stability with an acceptable drop in mass after wheat straw integration, whereas DSC plots show an increase in peak temperature as well as enthalpy, which increases the thermal capacity of the compounds in which wheat straw was integrated. Wheat straw fibers induced an increase in pores and an acceptable distribution in the plaster matrix, resulting in excellent adhesion between the two compounds, according to SEM pictures.

2. As for the thermophysical analytics, the samples were prepared, with different percentages of wheat straw, were prepared and tested to investigate their hygrothermal behavior. The results achieved from the test measurements show that the addition of wheat straw in the plaster matrix resulted in a linear reduction in the density and an increase in porosity. This means a reduction in thermal conductivity and therefore a more insulating behavior of the material. Furthermore, The time lag of a wall with varied configurations is impacted by the thermophysical parameters of the wall's material, thickness, and orientation. So there is an inverse connection between the time lag and that of thermal energy and thermal diffusion. This is because as the wall thickness gets thicker, its heat storage capacity increases. Plaster containing 15% wheat straw has good energy saving, i.e., a small thermal conductivity.

The findings revealed that the physical and thermal characteristics of plaster might be enhanced. As a result, the newly created composite may function as an efficient replacement for standard plaster materials that meet the building criteria.

**Author Contributions:** L.S.A.: Conceptualization, methodology, formal analysis, investigation, writing—original draft preparation. N.B.: Conceptualization, writing—review, editing, supervision. All authors have read and agreed to the published version of the manuscript.

**Funding:** This research received no external funding.

**Acknowledgments:** The authors wish to express their sincere gratitude to the General Directorate for Scientific Research and Technological Development (DGRSDT), Alger.

**Conflicts of Interest:** The authors declare no conflict of interest.

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
