# Peer review of "Thermal and Physico-Chemical Characteristics of Plaster Reinforced with Wheat Straw for Use as Insulating Materials in Building"

_buildings, doi:10.3390/buildings12081119_

Round 1

Reviewer 1 Report

Dear authors,

you performed an overall interesting study on the addition of wheat straw to a gypsum render. I see the merit mostly in the detailed characeterisation of the materials. However, the explanation of this characterisation is not always clear and some of the conclusions are not merited in my opinion. If these shortcomings can be addressed, I believe this could be a good paper.

I have tried to explain some of the issues in the following:

The manuscript needs strong English language editing (e.g. capitalisations, "comerciale" instead of commercial etc.); tenses are used alternatingly in the methods.

Introduction: There some inconsistencies in the introduction, e.g. it is stated: "The manufacture of cement requires more energy than the production of gypsum [6], so gypsum and plaster are used in construction because of their characteristics and ease of production [7]." However, there are also plasters made from cement. One should use the term stone wool not rock wool (which is a commercial product).

The conclusion that the addition of wheat leads to pore formation in the gypsum matrix is based only on four SEM images which were made at different magnification. I do not think this is conclusive. In order to make such a conclusion, pore size distribution of the gypsum matrix needs be studied in more detail, e.g. using BET or maybe more SEM images.

l.6: Which "vehicles"?

l.52/l.229: Sections should be made according to the typical standards and labelled correctly. I.e. Section 2 should be called "Materials and experimental methods", Section 5 should be part of the results.

l.56: size of the pieces should be mentioned

l.61/62: it should be clarified if the proportions are wrt. weight or volume; the same goes for l.69/70

l.88: X-ray diffraction is typically abbreviated "XRD" not "DRX"

l.111ff: not clear what kind of device was used; CT-meters are typically current transformer meters which does not make sense here; ref. 17 seems not to be publicly available, so more information cannot be found; the procedure should either be explained better (maybe also with an illustration) or made more concise since the explanation of the procedure give cannot be followed without knowing it already. It is not stated how samples were conditioned before measurement.

l.113ff: The procedure is not clear at all; which inside and outside surfaces are meant? (probably of a building but there is no mention of a demonstrator); it is not clear what apparatus is used etc.

l.131ff: Definitions of heat flux are inconsistent (eq. 6-8): Phi_plaster is defined in two different ways.

l.164: What is the criterion for being "acceptable"?

Fig.2-4: labels for the samples are inconsistent

Fig.4/l.176: The curve for pure plaster varies from the composites.

l.190: What is "net plaster"?

l.217: write SEM instead of MEB

Fig.10: images should be taken at comparable magnification

Fig.14 & 15: Individual samples should be indicated by marks in the plots

l.272/274: Such a statement does not hold in general. It depends on the outside and inside temperatures, on the insulation/wall thickness as well as on the day and night temperature differences of the outside of the building.

l.289/290: This comparison to the performance in ref. 8 is not correct. First of all, a thermal conductivity of about 310 mW/(m.K) is given for 25% straw in Fig.15 of ref. 8. Furthermore, the composition of the samples in the reference are very different from the ones prepared here which should be mentioned in a comparison.

Sec. 5.4: The comparison to the references in this section is not convincing since the materials and hence the base scenario is different.

The conclusions are not all convincing, for example the effect on the porosity of the material is not supported clearly and the time lag is only increased slightly with the addition of the fibres.

Author Response

you will find the response to your comment in the file bellow

Reviewer 2 Report

1. Please check the case sensitivity and polish further the grammars in the whole manuscript, including but not limited to the following:

1) One of the ways to reduce energy in buildings is to improve the insulation properties of building envelopes [5], Insulation is done by treating natural materials (rock wool) or using materials extracted from petrochemicals (polystyrene).

2) Despite this, its environmental impact is very significant [6].

3) Knowing the bulk density is a necessary process in such studies.

4) The difference, however, is in the percentage of weight loss, which changes depending on the percentage of wheat straw added to the plaster, so that the greater the percentage of wheat straw added in the samples The weight loss was greater compared to the plaster sample.

5) The third region, which is characterized by a temperature range of 360 to 650 ℃, so the weight loss in this area is related to the chemical bonding of water with hydraulic compounds in relation to plaster and to the deterioration of carbon in relation to wheat straw.

6) On the other hand, and through the figure (13) and the results in the table (2), thermal diffusivity decreased from 3.4107 to 2.87107 m2.s1, which is an estimated rate of decrease of 15.6% in heat transfer, The volumetric heat capacity also decreased by 5.84%.

7) As the thickness of the samples increases, so does the time lag value, and this can be explained by the fact that in the wall, which bears a small thickness, the heat wave spreads from outside to inside of wall without any delay, however, if the wall was made of insulating materials with a low heat capacity and thermal conductivity, the value of time lag increases, which is confirmed by [50], who studied such cases, so that the time lag values.

2. The author attributes the exothermic peak between 350 and 370 °C to the thermal decomposition of wheat straw, whereas it is also detected in the pure CP without wheat straw. The intensity of the exothermic peak between 350 and 370 °C shows vast difference with the filler contents increased. Please explain why these happen and improve the analysis in this paper.

3. I suggest the DRX and FTIR of pure plaster and the composites to be displayed and compared in the same graph.

4. Please supplement the marks for FTIR peaks of net fiber in figure (7-a) refer to that of CP in figure (7-b).

5. How can the author analyze the number and size of pore according to figure 10 without making the scale of SEM images consistent? And the order of "15%", "0%", "10%" and "5%" in figure 10 doesn’t seems correct to me. Please explain and renovate the SEM images.

6. Please add the error bar to the results in figure 11-17.

Author Response

Thank you for your efforts. With this message, you will find a file with sincere answers to all your questions and suggestions

Reviewer 3 Report

The paper "Thermal and physico-chemical characteristics of plaster reinforced with wheat straw for use as insulating materials in building" falls within the scope of the Buildings journal and shows technical relevance.

In this paper the authors present an interesting comparison between four different dosages of gypsum compounds three of them including different percentages of wheat straw for the analysis of their performance as thermal insulating materials.

The paper is appropriately structured and presented and the material is publishable but requires improvement. In this sense, some suggestions on the attached paper should be addressed before publishing.

Suggestion 01

Considering the wide range of research on this topic in the literature, a more in-depth review of what has been published in recent years is lacking. Furthermore, in the introduction section, novelty is unclear: What is the original contribution of the study? This section is not very enlightening on the subject. Novelty should be made as clear as possible.

Suggestion 02

The limitations of the study should also be included.

Suggestion 03

A synthetic diagram outlining the methodology followed would improve readability.  It is advisable to introduce it at the end of the Introduction section, as well as a paragraph summarising phases and links

Suggestion 04

Conclusion section needs to be also improved. Please paraphrase your results and discussions and use them in the conclusion part.

Author Response

(The authors gave the same response as above.)

Round 2

Reviewer 2 Report

All my concerns were addressed in the revised version. I believe the manuscript  can be accepted in its current state.

Reviewer 3 Report

Since the authors have responded adequately to all comments made on the initial version of the paper, this reviewer has no further objection to its publication in the present form.